# Maximizing Ensemble Diversity in Deep Reinforcement Learning

**Hassam Ullah Sheikh** [*]
Intel Labs
hassam.sheikh@intel.com

**Mariano Phielipp**
Intel Labs
mariano.j.phielipp@intel.com

**Ladislau Bölöni**
Department of Computer Science
University of Central Florida
lboloni@cs.ucf.edu

## Abstract

Modern deep reinforcement learning (DRL) has been successful in solving a range of challenging sequential decision-making problems. Most of these algorithms use an ensemble of neural networks as their backbone structure and benefit from the diversity among the neural networks to achieve optimal results. Unfortunately, the members of the ensemble can converge to the same point either the parametric space or representation space during the training phase, therefore, losing all the leverage of an ensemble. In this paper, we describe Maximize Ensemble Diversity in Reinforcement Learning (MED-RL), a set of regularization methods inspired from the economics and consensus optimization to improve diversity in the ensemble-based deep reinforcement learning methods by encouraging inequality between the networks during training. We integrated MED-RL in five of the most common ensemble-based deep RL algorithms for both continuous and discrete control tasks and evaluated on six Mujoco environments and six Atari games. Our results show that MED-RL augmented algorithms outperform their un-regularized counterparts significantly and in some cases achieved more than 300% in performance gains.

## 1 Introduction

Reinforcement learning (RL) agents trained with high capacity function approximators such a deep neural networks have shown to solve complex sequential decision-making problems, including the board games of Chess, GO and Shogi (Silver et al., 2016; 2017; 2018), achieving super-human performance in video games (Mnih et al., 2015; Vinyals et al., 2019) and solving robotic manipulation tasks (Liu et al., 2021). Despite achieving these tremendous goals, modern deep reinforcement learning (DRL) algorithms have plethora of limitations. For example, it is well-known that DRL algorithms are sample-inefficient and require stupendous amount of environment interactions to learn an optimal policy (Łukasz Kaiser et al., 2020). Additional problems encountered and exacerbates during training a DRL agent includes the overestimation bias that occurs while estimating the target values for Q-learning (Fujimoto et al., 2018; Lan et al., 2020; Hado van Hasselt et al., 2016), error propagation during Bellman backup (Kumar et al., 2019) and trade-off between exploration and exploitation (Chen et al., 2017).

Recently, the use of ensemble has been a popular choice to address the above mentioned issues. These methods combine multiple neural networks to model the value functions or (and) the policy (Osband et al., 2016; Chen et al., 2017; Lan et al., 2020; Lee et al., 2020). For example, TD3 (Fujimoto et al., 2018) used two critics to address the overestimation bias problem in continuous control problems while MaxminDQN (Lan et al., 2020) provided a mechanism to use the cardinality of the ensemble to use as a knob to tune between over and under estimation in deep Q-learning. Similarly, Bootstrapped DQN (Osband et al., 2016; Chen et al., 2017) used ensemble for effective exploration.

---

[*]Partial work done while being a Ph.D student at University of Central Florida

The primary insight of this paper is that the performance of ensemble based methods is contingent on maintaining sufficient *diversity* between the neural networks of the ensemble. If the neural networks in the ensembles converge to a common representation (we will show that this is the case in many scenarios), the performance of these approaches significantly degrades. We note that even with different representations, the Q-values will still converge towards a shared optimum, but they are statistically less likely to follow the same learning trajectory elsewhere.

In this paper, we propose **M**aximize **E**nsemble **D**iversity in **R**einforcement **L**earning (MED-RL), a set of regularization methods inspired from the economics and consensus optimization to improve diversity and to prevent the collapse of the representations in the ensemble-based deep reinforcement learning methods by encouraging inequality between the networks during training. The objective of the regularizers is solely to keep the representations different, while still allowing the models to converge to the optimal Q-value. The motivation for the regularizers came from topic of income distribution in economic theory that provides a rich source of mathematical formulations that measure inequality. While in economics, high inequality is seen as a negative, in our case we used the inequality metrics to encourage diversity between the neural networks.

To summarize, our contributions are following:

1. We *empirically* show that high representation similarity between neural network based Q-functions leads to degradation in performance in ensemble based Q-learning methods.

2. To mitigate this, we propose five regularizers based on inequality measures from economics theory and consensus optimization that maximize diversity between the neural networks in ensemble based reinforcement learning methods.

3. We integrated MED-RL in TD3 (Fujimoto et al., 2018), SAC (Haarnoja et al., 2018) and REDQ (Chen et al., 2021) for continuous control tasks and in MaxminDQN (Lan et al., 2020) and EnsembleDQN (Anschel et al., 2017) for discrete control tasks and evaluated on six Mujoco environments and six Atari games. Our results show that MED-RL augmented algorithms outperform their un-regularized counterparts significantly and in some cases achieved more than 300% in performance gains and are up to 75% more sample-efficient.

4. We also show that MED-RL augmented SAC is more sample-efficient than REDQ, an ensemble based method specifically designed for sample-efficiency, and can achieve similar performance to REDQ up to 50 times faster on wall-clock time.

## 2 RELATED WORK

**Ensembles in Deep RL:**  Use of an ensemble of neural networks in Deep RL has been studied in several recent studies for different purposes. In (Fujimoto et al., 2018; Anschel et al., 2017; Lan et al., 2020) have used an ensemble to address the overestimation bias in deep Q-learning based methods for both continuous and discrete control tasks. Similarly, Bootstrapped DQN and extensions (Osband et al., 2016; Chen et al., 2017) have leveraged ensemble of neural networks for efficient exploration. The problem of error propagation in Bellman backup was addressed in (Kumar et al., 2019) using an ensemble of neural networks. Sample efficiency, a notorious problem in RL has taken advantage from an ensemble (Chen et al., 2021). Recently, SUNRISE (Lee et al., 2020) proposed a unified framework for ensemble-based deep reinforcement learning.

**Diversity in Ensembles:**  Diversity in neural network ensembles has been studied years before the resurgence of deep learning (Brown, 2004). Even though diversity is an important topic in neural networks, most of the studies in this topic revolve around addressing problems in supervised learning settings. More recently there has been a number of studies that have diversity in ensembles to measure and improve model uncertainty. Jain et al. (2020) have proposed a diversity regularizer to improve the uncertainty estimates in out-of-data distribution. Lee et al. (2015) have used Multiple choice Learning to learn diverse Convolutional Neural Networks for image recognition.

**Regularization in Reinforcement Learning:**  Regularization in reinforcement learning has been used to perform effective exploration and learning generalized policies. For instance, (Grau-Moya et al., 2019) uses mutual-information regularization to optimize a prior action distribution for better performance and exploration, (Cheng et al., 2019) regularizes the policy $\pi(a|s)$ using a control

prior, (Galashov et al., 2019) uses temporal difference error regularization to reduce variance in Generalized Advantage Estimation (Schulman et al., 2016). Generalization in reinforcement learning refers to the performance of the policy on different environment compared to the training environment. For example, (Farebrother et al., 2018) studied the effect of $L^2$ norm on DQN on generalization, (Tobin et al., 2017) studied generalization between simulations vs. the real world, (Pattanaik et al., 2018) studied parameter variations and (Zhang et al., 2018) studied the effect of different random seeds in environment generation.

**Diversity in Reinforcement Learning:**   Diversity in reinforcement learning is active area of research. (Pacchiano et al., 2020) uses Determinantal Point Processes to promote behavioral diversity, Lupu et al. (2021) have used policy diversity to improve zero-shot coordination in multi-agent setting. (Tang et al., 2021) uses reward randomization for discovering diverse strategic policies in complex multi-agent games. In (Li et al., 2021) proposed CDS that uses information-theoretical objective to maximize the mutual information between agents' identities and trajectories and encourage diversity. More recently (An et al., 2021) have used diversified Q-ensembles to address overestimation in offline reinforcement learning.

**Representation Similarity:**   Measuring similarity between the representations learned by different neural networks is an active area of research. For instance, (Raghu et al., 2017) used Canonical Correlation Analysis (CCA) to measure the representation similarity. CCA find two basis matrices such that when original matrices are projected on these bases, the correlation is maximized. (Raghu et al., 2017; Mroueh et al., 2015) used truncated singular value decomposition on the activations to make it robust for perturbations. Other work such as (Li et al., 2015) and (Wang et al., 2018) studied the correlation between the neurons in the neural networks.

## 3  BACKGROUND

**Reinforcement learning:**   We consider an agent as a Markov Decision Process (MDP) defined as a five element tuple $(\mathcal{S}, \mathcal{A}, P, r, \gamma)$, where $\mathcal{S}$ is the state space, $\mathcal{A}$ is the action space, $P : \mathcal{S} \times \mathcal{A} \times \mathcal{S} \rightarrow [0, 1]$ are the state-action transition probabilities, $r : \mathcal{S} \times \mathcal{A} \times \mathcal{S} \rightarrow \mathbb{R}$ is the reward mapping and $\gamma \rightarrow [0, 1]$ is the discount factor. At each time step $t$ the agent observes the state of the environment $s_t \in \mathcal{S}$ and selects an action $a_t \in \mathcal{A}$. The effect of the action triggers a transition to a new state $s_{t+1} \in \mathcal{S}$ according to the transition probabilities $P$, while the agent receives a scalar reward $R_t = r(s_t, a_t, s_{t+1})$. The goal of the agent is to learn a policy $\pi$ that maximizes the expectation of the discounted sum of future rewards.

**Representation Similarity Measure:**   Let $X \in \mathbb{R}^{n \times p_1}$ denote a matrix of activations of $p_1$ neurons for $n$ examples and $Y \in \mathbb{R}^{n \times p_2}$ denote a matrix of activations of $p_2$ neurons for the same $n$ examples. Furthermore, we consider $K_{ij} = k(x_i, x_j)$ and $L_{ij} = l(y_i, y_j)$ where $k$ and $l$ are two kernels.

Centered Kernel Alignment (CKA) (Kornblith et al., 2019; Cortes et al., 2012; Cristianini et al., 2002) is a method for comparing representations of neural networks, and identifying correspondences between layers, not only in the same network but also on different neural network architectures. CKA is a normalized form of Hilbert-Schmidt Independence Criterion (HSIC) (Gretton et al., 2005). Formally, CKA is defined as:

$$\text{CKA}(K, L) = \frac{\text{HSIC}(K, L)}{\sqrt{\text{HSIC}(K, K) \cdot \text{HSIC}(L, L)}}$$

HSIC is a test statistic for determining whether two sets of variables are independent. The empirical estimator of HSIC is defined as:

$$\text{HSIC}(K, L) = \frac{1}{(n-1)^2} \text{tr}(KHLH)$$

where $H$ is the centering matrix $H_n = I_n - \frac{1}{n}\mathbf{1}\mathbf{1}^T$.

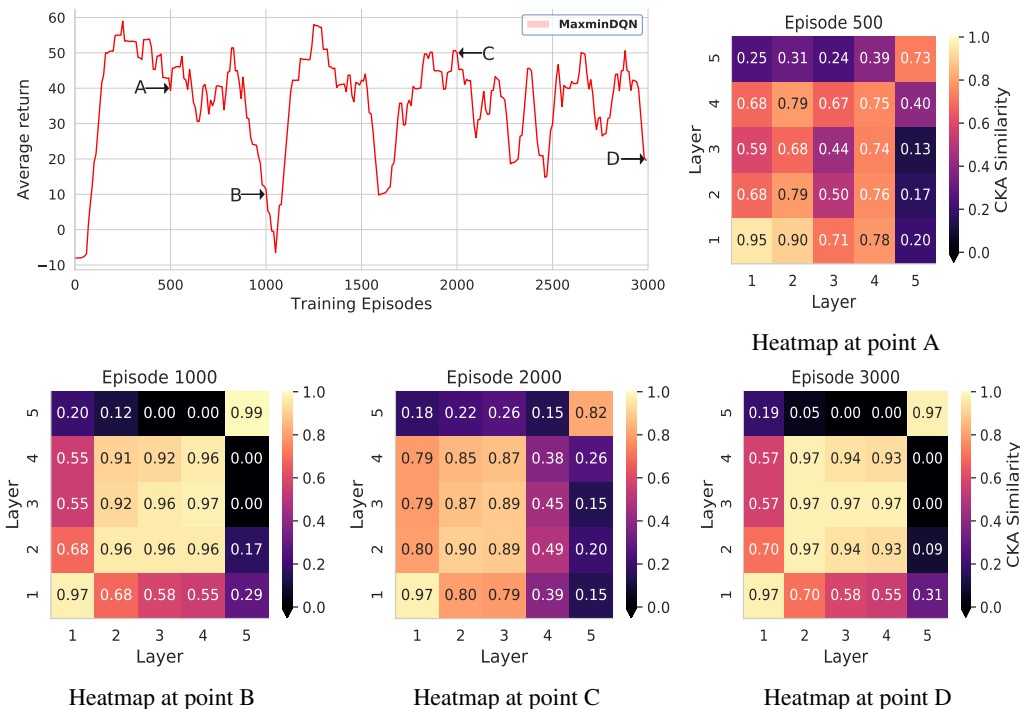

Figure 1: The training graph and CKA similarity heatmaps of a MaxminDQN agent with 2 neural networks. The letters on the plot show the time when CKA similarities were calculated. Heatmaps at A and C have relatively low CKA similarity and have relatively higher average return as compared to heatmaps at point B and D that have extremely high similarity across all the layers. *See diagonal values from bottom left to top right.*

## 4 MAXIMIZE ENSEMBLE DIVERSITY IN REINFORCEMENT LEARNING

In this section, we propose MED-RL: **M**aximize **E**nsemble **D**iversity in **R**einforcement **L**earning, a set of regularizers inspired from the Economics and consensus optimization to improve diversity and to prevent the collapse of the representations in the ensemble-based deep reinforcement learning methods by encouraging inequality between the networks during training. This section is organized as follows:

1. We *empirically* show that high representation similarity between neural network based Q-functions leads to degradation in performance in ensemble based Q-learning methods.

2. we present the Economics theory and consensus optimization inspired regularizers with their mathematical formulation.

### 4.1 EMPIRICAL EVIDENCE TO CORRELATE PERFORMANCE AND REPRESENTATION SIMILARITY

The work in this paper starts from the conjecture that high representation similarity between neural networks in an ensemble-based Q-learning technique correlates to poor performance. To empirically verify our hypothesis, we trained a MaxminDQN (Lan et al., 2020) agent with two neural networks on the Catcher environment (Qingfeng, 2019) for about 3000 episodes ($5 \times 10^6$ training steps) and calculated the CKA similarity with a linear kernel after every 500 episodes. The training graph along with the CKA similarity heatmaps are shown in Figure 1. Notably at episode 500 (heatmap A) and episode 2000 (heatmap C), the representation similarity between neural networks is low but the average return is relatively high. In contrast, at episode 1000 (heatmap B) and episode 3000 (heatmap D) the representation similarity is highest but the average return is lowest.

## 4.2 REGULARIZATION FOR MAXIMIZING ENSEMBLE DIVERSITY

In order to maximize the ensemble diversity, we propose to regularize the training algorithm with an additional criteria that favors diversity between the ensembles. In the following, $N$ is the number of neural networks in the ensemble, $\ell_i$ is the $L^2$ norm of the $i$-th neural network's parameters, $\bar{\ell}$ is the mean of all the $L^2$ norms and $\ell$ is the list of all the $L^2$ norms.

The first four metrics we consider are based on inequality measures from economic theory. While in economics, inequality is usually considered something to be avoided, in our case we aim to increase inequality (and thus, ensemble diversity).

The **Atkinson Index** (Atkinson et al., 1970) measures income inequality and is useful in identifying the end of the distribution that contributes the most towards the observed inequality. Formally, it is defined as

$$A_\epsilon = \begin{cases} 1 - \dfrac{1}{\bar{\ell}}\left(\dfrac{1}{N}\sum_{i=1}^{N}\ell_i^{1-\epsilon}\right)^{\frac{1}{1-\epsilon_{at}}}, & \text{for } 0 \le \epsilon_{at} \neq 1, \\[3ex] 1 - \dfrac{1}{\bar{\ell}}\left(\dfrac{1}{N}\prod_{i=1}^{N}\ell_i\right)^{\frac{1}{N}}, & \text{for } \epsilon_{at} = 1, \end{cases} \tag{1}$$

where $\epsilon_{at}$ is the inequality aversion parameter used to tune the sensitivity of the measured change. When $\epsilon_{at} = 0$, the index is more sensitive to the changes at the upper end of the distribution, while it becomes sensitive towards the change at the lower end of the distribution when $\epsilon_{at}$ approaches 1.

The **Gini coefficient** (Allison, 1978) is a statistical measure of the wealth distribution or income inequality among a population and defined as the half of the relative mean absolute difference:

$$G = \frac{\sum_{i=1}^{N}\sum_{j=1}^{N}|\ell_i - \ell_j|}{2N^2\bar{\ell}} \tag{2}$$

The Gini coefficient is more sensitive to deviation around the middle of the distribution than at the upper or lower part of the distribution.

The **Theil index** (Johnston, 1969) measures redundancy, lack of diversity, isolation, segregation and income inequality among a population. Using the Theil index is identical to measuring the redundancy in information theory, defined as the maximum possible entropy of the data minus the observed entropy:

$$T_T = \frac{1}{N}\sum_{i=1}^{N}\frac{\ell_i}{\bar{\ell}}\ln\frac{\ell_i}{\bar{\ell}} \tag{3}$$

The **variance of logarithms** (Ok & Foster, 1997) is a widely used measure of dispersion with natural links to wage distribution models. Formally, it is defined as:

$$V_L(\ell) = \frac{1}{N}\sum_{i=1}^{N}[\ln\ell_i - \ln g(\ell)]^2 \tag{4}$$

where g($\ell$) is the geometric mean of $\ell$ defined as $(\prod_{i=1}^{N}\ell_i)^{1/N}$.

The final regularization method we use is inspired from consensus optimization. In a consensus method (Boyd et al., 2011), a number of models are independently optimized with their own task-specific parameters, and the tasks communicate via a penalty that encourages all the individual solutions to converge around a common value. Formally, it is defined as

$$M = \|\bar{\theta} - \theta_i\|^2 \tag{5}$$

Where $\bar{\theta}$ is the mean of the parameters of all the neural networks and $\theta_i$ represents the parameters of the $i - th$ neural network. We will refer this regularizer as MeanVector throughout this paper. For completeness, the algorithm shown in Algorithm 1. Notice that the regularization term appears with a negative sign, as the regularizers are essentially inequality metrics that we want to maximize.

### 4.3 TRAINING ALGORITHM

Using the regularization functions defined above, we can develop diversity-regularized variants of the the ensemble based algorithms. The training technique is identical to the algorithms described in (Lan et al., 2020; Anschel et al., 2017; Fujimoto et al., 2018; Haarnoja et al., 2018; Chen et al., 2021), with a regularization term added to the loss of the Q-functions. The loss term for $i$-th Q-function with parameters $\psi_i$ is:

$$\mathcal{L}\left(\psi_i\right) = \mathbb{E}_{s,a,r,s'}\left[\left(Q_\psi^i\left(s,a\right) - Y\right)^2\right] - \lambda \mathcal{I}\left(\ell_i, \boldsymbol{\ell}\right),$$

where $Y$ is the target value depending on the algorithm, $\mathcal{I}$ is the regularizer of choice from the list above and $\lambda$ is the regularization weight. Notice that the regularization term appears with a negative sign, as the regularizers are essentially inequality metrics that we want to maximize. As a reference the modified algorithm for MaxminDQN is shown in Algorithm 1.

## 5 EXPERIMENTS

### 5.1 ISN'T RESAMPLING AND DIFFERENT INITIALIZATION OF WEIGHTS ENOUGH?

The most common question that comes to mind to address the diversity issue is why not initialize the neural networks with different weights and train each network with a different sample from the buffer? This approach has been thoroughly discussed in (Brown, 2004) and have been shown to be ineffective. To re-iterate the findings in (Brown, 2004), we performed a regression experiment in which we learnt a *sine* function using two different three layered fully connected neural networks with 64 and 32 neurons in each hidden layer with ReLU. The neural networks were *initialized using different weights* and were trained using different batch sizes $(512, 128)$ and learning rates $(1e^{-4}, 1e^{-3})$. The Figure 2a shows the learnt functions while Figure 2b represents their CKA similarity heatmap before and after training. The odd numbered layers represent pre-ReLU activations while the even numbered layers represent post-ReLU activations. It can be seen that before training, the CKA similarity between the two neural networks from layer 4 and onward is relatively low and the output being 0% similar while after training, the trained networks have learnt highly similar representation while their output being 98% similar.

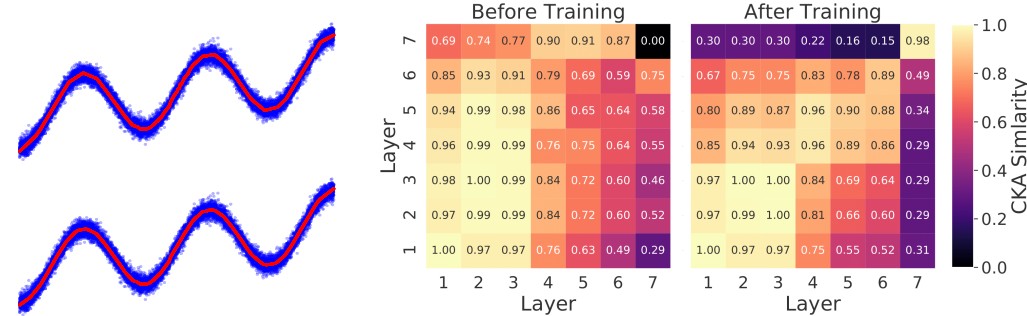

(a) Regression using two different neural networks

(b) CKA similarity heatmap between different layers of the two neural networks used for the regression experiment.

Figure 2: **Left:** Fitting a sine function using two different neural network architectures. The upper function was approximated using 64 neurons in each hidden layer while the lower function used 32 neurons in each hidden layer. **Right:** Represents the CKA similarity heatmap between different layers of both neural networks before and after training. The right diagonal (bottom left to top right) measures representation similarity of the corresponding layers of both neural networks. The trained networks have learnt similar representations while their output was 98% similar. *See diagonal values from bottom left to top right.*

This example shows that neural networks can learn similar representation while trained on different batches. This observation is important because in MaxminDQN and EnsembleDQN training,

each neural network is trained on a separate batch from the replay buffer but still learns similar representations.

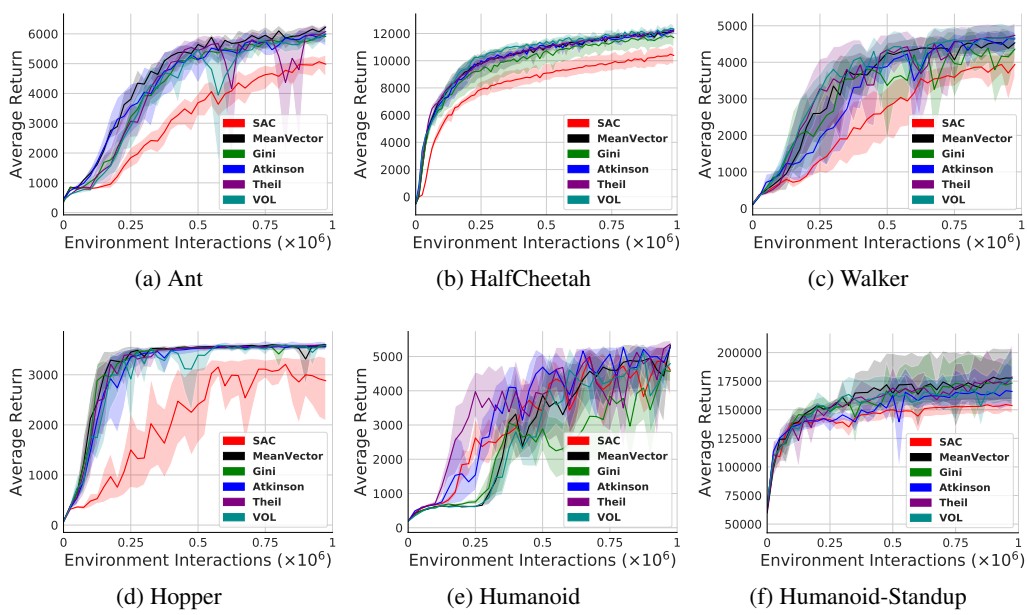

Figure 3: Training curves and 95% confidence interval (shaded area) for the MED-RL augmented variants for SAC

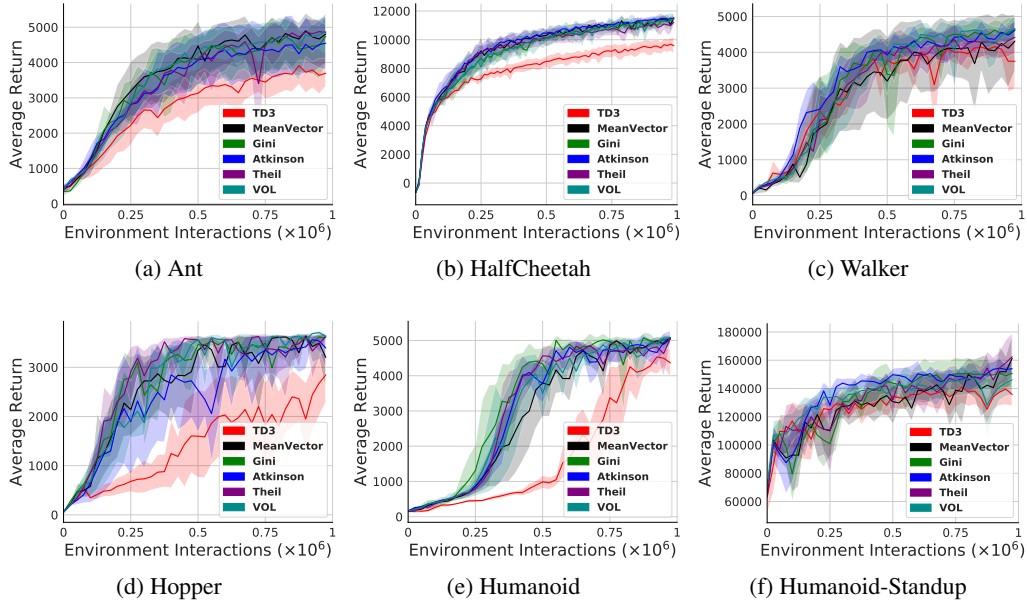

Figure 4: Training curves and 95% confidence interval (shaded area) for the MED-RL augmented variants for TD3

## 5.2 EXPERIMENTAL SETUP

**Continuous control tasks:** We evaluated MED-RL augmented continuous control algorithms such as TD3, SAC and REDQ on the Mujoco continuous control benchmark environments. We compared the results of MED-RL with un-regularized counterparts. We report the mean and standard deviation across five runs after 1M timesteps on six complex environments: Cheetah, Walker, Hopper, Ant, Humanoid and Humanoid-Standup. For REDQ, we evaluated on Cheetah, Walker, Hopper and Ant for 300K timesteps only.

## 5.3 COMPARATIVE EVALUATION

**Continuous control tasks:** Tables 1 to 3 show the average returns of evaluation roll-outs for all the continuous control methods. MED-RL consistently improves the performance of SAC, TD3 and REDQ in all the experiments. Even though the focus of this work is to maximize the average return, we find that MED-RL augmented algorithms are more sample-efficient than their un-regularized counterparts (see Figures 3, 4 and 5). For example, it can be seen Figure 3b that baseline SAC reaches the average return of 10K in about 1M environment interactions while all MED-RL variants reach the same average return in approximately 250K environment interaction, therefore, being approximately 75% more sample efficient than the baseline SAC. This improvement in sample-efficiency can be noted in all nearly all the experiments except SAC-Humanoid and TD3-Walker experiments. The training plots for REDQ are shown in Appendix.

Table 1: Max Average Return for MED-RL SAC over 5 trials of 1 million time steps. Maximum value for each task is bolded. ± corresponds to a single standard deviation over trials

| Environment | Baseline | MeanVector | Gini | Atkinson | Theil | VOL |
|---|---|---|---|---|---|---|
| HalfCheetah | $10380.3 \pm 681.8$ | $12278.1 \pm 160.3$ | $11691.2 \pm 715.6$ | $12117.2 \pm 304.7$ | $12212.7 \pm 216.7$ | $\mathbf{12339.5 \pm 284.9}$ |
| Ant | $4802.4 \pm 605.1$ | $\mathbf{6298.7 \pm 101.8}$ | $6047.8 \pm 167.4$ | $6163.2 \pm 207.6$ | $6091.5 \pm 222.3$ | $5965.1 \pm 196.4$ |
| Hopper | $2882.5 \pm 738.3$ | $\mathbf{3604.3 \pm 27.8}$ | $3552.9 \pm 60.5$ | $3560.4 \pm 82.2$ | $3596.6 \pm 57.7$ | $3587.7 \pm 42.9$ |
| Walker2d | $3954.9 \pm 356.7$ | $4525.7 \pm 340.9$ | $4523.0 \pm 440.1$ | $4659.8 \pm 253.0$ | $\mathbf{4753.8 \pm 394.7}$ | $4653.4 \pm 391.2$ |
| Humanoid | $4582.2 \pm 592.4$ | $\mathbf{5359.1 \pm 42.0}$ | $5224.6 \pm 105.1$ | $5275.1 \pm 40.4$ | $5355.2 \pm 137.3$ | $5311.7 \pm 49.1$ |
| Humanoid-Standup | $153633.2 \pm 8256.6$ | $177666.5 \pm 30044.1$ | $170592.6 \pm 29346.3$ | $164967.6 \pm 19464.6$ | $\mathbf{180268.1 \pm 33080.4}$ | $179645.1 \pm 29980.4$ |

Table 2: Max Average Return for MED-RL TD3 over 5 trials of 1 million time steps. Maximum value for each task is bolded. ± corresponds to a single standard deviation over trials

| Environment | Baseline | MeanVector | Gini | Atkinson | Theil | VOL |
|---|---|---|---|---|---|---|
| HalfCheetah | $9583.1 \pm 682.6$ | $\mathbf{11539.4 \pm 278.1}$ | $11477.9 \pm 405.2$ | $11442.5 \pm 187.8$ | $11232.7 \pm 323.6$ | $11393.6 \pm 532.7$ |
| Ant | $3829.1 \pm 675.7$ | $4829.6 \pm 1036.9$ | $4611.5 \pm 781.9$ | $4565.7 \pm 908.4$ | $4810.7 \pm 347.9$ | $\mathbf{4881.1 \pm 831.6}$ |
| Hopper | $2965.3 \pm 423.5$ | $\mathbf{3651.3 \pm 57.7}$ | $3629.5 \pm 92.8$ | $3582.3 \pm 153.9$ | $3649.1 \pm 62.7$ | $3614.1 \pm 89.2$ |
| Walker2d | $4140.6 \pm 334.2$ | $4396.3 \pm 837.5$ | $\mathbf{4666.3 \pm 319.7}$ | $4652.7 \pm 310.0$ | $4528.5 \pm 507.1$ | $4630.0 \pm 405.1$ |
| Humanoid | $4347.4 \pm 456.2$ | $5060.5 \pm 127.4$ | $5048.8 \pm 199.3$ | $\mathbf{5116.3 \pm 278.5}$ | $5096.1 \pm 98.1$ | $5040.0 \pm 112.7$ |
| Humanoid-Standup | $135176.6 \pm 7991.2$ | $160293.8 \pm 19657.2$ | $151123.1 \pm 12712.8$ | $154652.2 \pm 5607.7$ | $\mathbf{160481.5 \pm 15229.6}$ | $146970.3 \pm 13199.6$ |

Table 3: Max Average Return for MED-RL REDQ over 5 trials of 300K time steps. Maximum value for each task is bolded. ± corresponds to a single standard deviation over trials

| Environment | Baseline | MeanVector | Gini | Atkinson | Theil | VOL |
|---|---|---|---|---|---|---|
| HalfCheetah | $8368.3 \pm 56.3$ | $10067.9 \pm 360.7$ | $\mathbf{10234.4 \pm 74.4}$ | $9926.9 \pm 319.0$ | $10161.6 \pm 461.7$ | $9664.8 \pm 1975.2$ |
| Ant | $3001.3 \pm 2083.5$ | $5446.8 \pm 186.7$ | $5801.7 \pm 42.3$ | $5616.2 \pm 86.3$ | $5885.6 \pm 181.0$ | $\mathbf{5897.4 \pm 16.7}$ |
| Hopper | $2876.9 \pm 584.7$ | $3477.3 \pm 43.6$ | $3565.9 \pm 40.9$ | $3524.8 \pm 2.8$ | $\mathbf{3596.6 \pm 72.1}$ | $3550.8 \pm 50.1$ |
| Walker2d | $3722.3 \pm 52.6$ | $4282.7 \pm 414.5$ | $4217.1 \pm 150.6$ | $4133.9 \pm 145.9$ | $\mathbf{5028.4 \pm 205.6}$ | $4249.2 \pm 201.3$ |

## 5.4 Sample Efficiency and Compute Time

Tables 1 to 3 show that MED-RL augmented continuous control algorithms outperform the baseline versions significantly and a visual inspection of Figures 3, 4 and 5 show that MED-RL augmented algorithms are more sample-efficient as well. But are they more sample-efficient than algorithms that are specifically designed for sample-efficiency such as REDQ? To answer this question, we took the bolded results from Table 1, referred as MED-RL in this section, and evaluated the number of environment interactions and wall-clock time it took for MED-RL to reach similar performance as that of baseline REDQ. As shown in Table 4, MED-RL achieves similar performance to REDQ in 50% and 20% few environment interactions on Ant and HalfCheetah environment respectively and have significantly surpassed REDQ on $300K$ environment interactions. MED-RL does not only improve sample-efficiency but significantly improves compute time. As shown in Table 4, MED-RL achieves similar performance to REDQ up to 50 times faster on wall-clock time. Note it can be argued that REDQ can be parallelized to achieve faster wall-clock time but here we are only comparing standard sequential implementations but that will not address the sample-efficiency issue.

Table 4: Comparison of MED-RL augmented SAC with baseline REDQ on sample-efficiency and wall-clock time.

|  | HalfCheetah | Ant | Hopper | Walker2d |
|---|---|---|---|---|
| REDQ average reward | 8368.6 ± 56.3 | 3001.3 ± 2083.5 | 2876.9 ± 584.7 | 3722.3 ± 52.6 |
| Environment interactions taken by MED-RL to reach REDQ performance | 232K ± 36.5K | 254K ± 24K | 152K ± 40.9K | 283.15K ± 77.8K |
| Wall clock time REDQ (in mins) | 1670.54 ± 188.66 | 1853.17 ± 66.26 | 1647.97 ± 202.57 | 1690.27 ± 245.1 |
| Wall clock time taken by MED-RL to reach REDQ performance (in mins) | 49.61 ± 9.25 | 43.81 ± 9.42 | 32.23 ± 10.3 | 65.2 ± 16.2 |
| MED-RL average reward after 300K environment interactions | 9369.23 ± 509.68 | 4095.38 ± 433.24 | 3482.74 ± 95.30 | 3487.751 ± 874.14 |
| Wall clock time taken by MED-RL for 300K environment interactions | 64.17 ± 9.71 | 51.29 ± 6.75 | 62.94 ± 6.71 | 69.44 ± 1.86 |

**Discrete control tasks:** We evaluated MED-RL augmented discrete control algorithms such as MaxminDQN and EnsembleDQN on the PyGames (Qingfeng, 2019) and MinAtar (Young & Tian, 2019). We chose these environments to have a fair comparison since we used the source code provided by MaxminDQN authors. We reused all the hyperparameter settings from (Lan et al., 2020) except the number of neural networks, which we limited to four and trained each solution for five fixed seeds. The results on the discrete control tasks are shown in the Appendix.

## 6 Conclusion

In this paper, we proposed Maximize Ensemble Diversity in Reinforcement Learning (MED-RL), a set of regularization methods inspired from the economics and consensus optimization to improve diversity in the ensemble-based deep reinforcement learning methods by encouraging inequality between the networks during training. We also empirically showed that high representation similarity between the networks of the ensemble could cause degradation in the performance. Our experiments have shown that MED-RL not only improves the average return of ensemble based reinforcement learning algorithms but can increase their sample-efficiency by approximately 75% when compared to their un-regularized counterparts. Additionally we have shown the SAC when augmented with MED-RL can outperform REDQ, an algorithm specifically designed for sample-efficiency, in both sample-efficiency and compute time.

**Acknowledgement:** This work had been supported in part by the National Science Foundation under grant number CNS-1932300

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

---

**Algorithm 1:** MED-RL: MaxminDQN version

---

The differences between the baseline MaxminDQN and MEDRL-MaxminDQN are highlighted

Initialize $N$ Q-functions $\{Q^1, \ldots, Q^N\}$ parameterized by $\{\psi_1, \ldots, \psi_N\}$

Initialize empty replay buffer $D$

Observe initial state $s$

**while** *Agent is interacting with the Environment* **do**

    $Q^{min}(s, a) \leftarrow \min_{k \in \{1, \ldots, N\}} Q^k(s, a), \forall a \in \mathcal{A}$

    Choose action $a$ by $\epsilon$-greedy based on $Q^{min}$

    Take action $a$, observe $r, s'$

    Store transition $(s, a, r, s')$ in $D$

    Select a subset $S$ from $\{1, \ldots, N\}$   (e.g., randomly select one $i$ to update)

    **for** $i \in S$ **do**

        Sample random mini-batch of transitions $(s_D, a_D, r_D, s'_D)$ from $D$

        Get update target: $Y^M \leftarrow r_D + \gamma \max_{a' \in A} Q^{min}(s'_D, a')$

        Generate list of $L^2$ norms : $\boldsymbol{\ell} = \left[ \|\psi_1\|^2, \ldots, \|\psi_N\|^2 \right]$

        Update $Q^i$ by minimizing $\mathbb{E}_{s_D, a_D, r_D, s'_D} \left( Q^i_{\psi_i}(s_D, a_D) - Y^M \right)^2 - \lambda \mathcal{I}(\ell_i, \boldsymbol{\ell})$

    **end**

    $s \leftarrow s'$

**end**

---

# A  TRAINING PLOTS OF REDQ

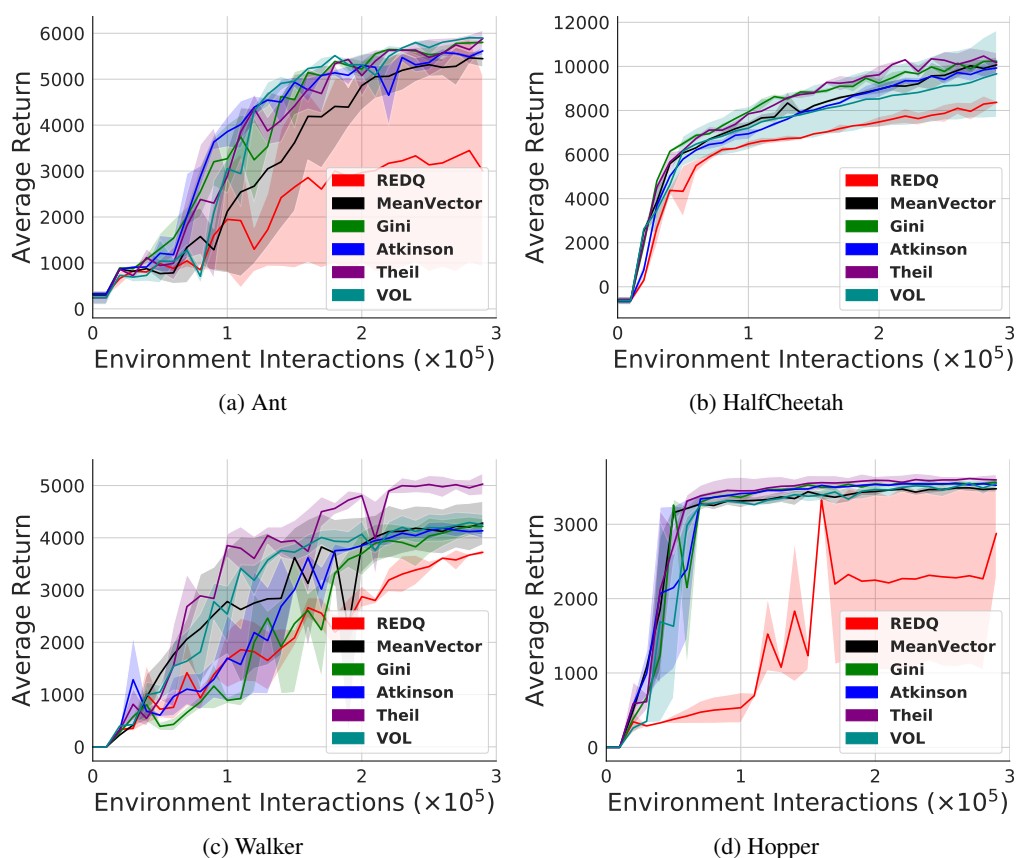

(a) Ant

(b) HalfCheetah

(c) Walker

(d) Hopper

Figure 5: Training curves and 95% confidence interval (shaded area) for the augmented variants for REDQ together with baseline REDQ.

# B    RESULTS ON DISCRETE CONTROL TASKS USING MAXMINDQN

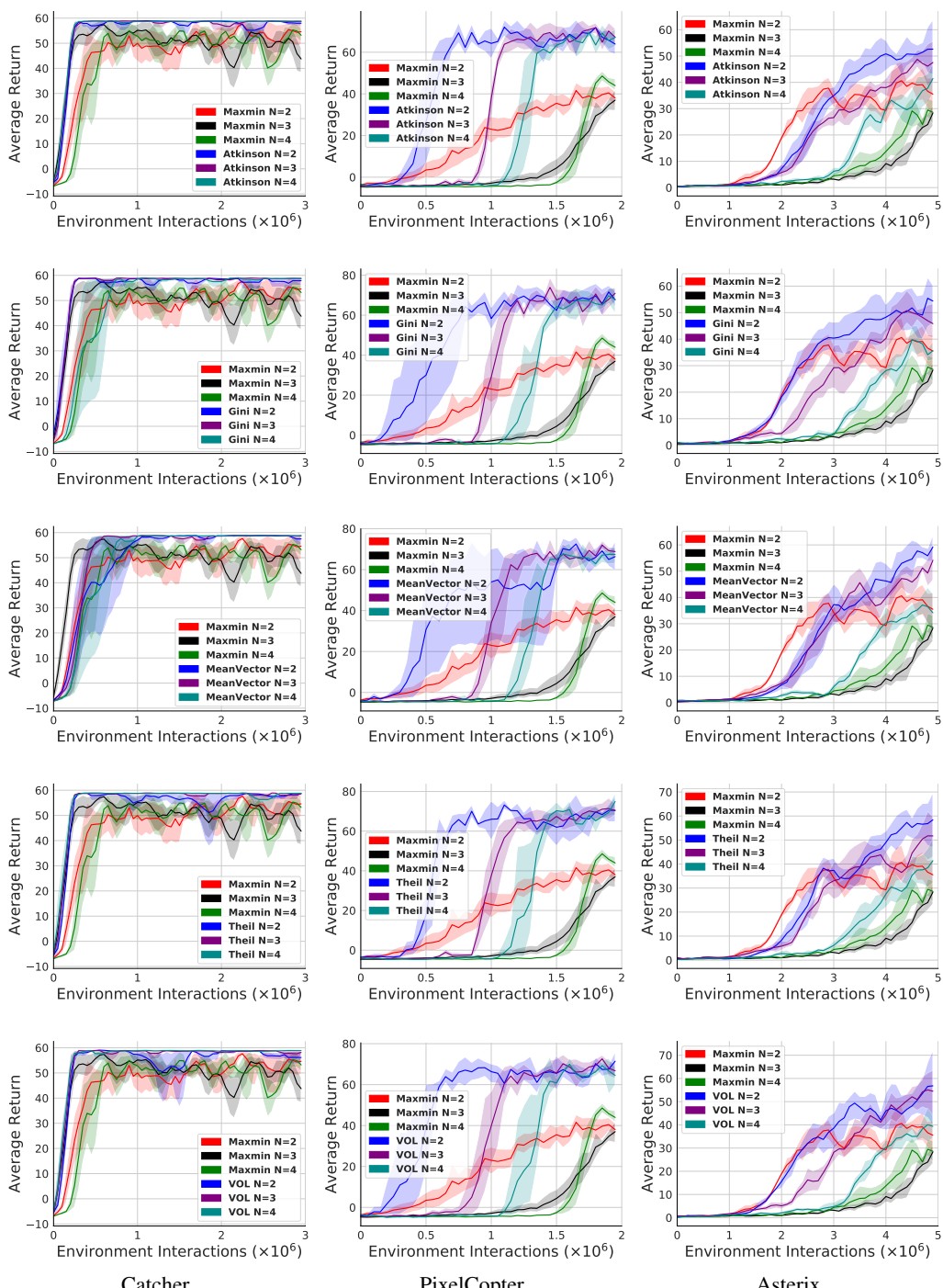

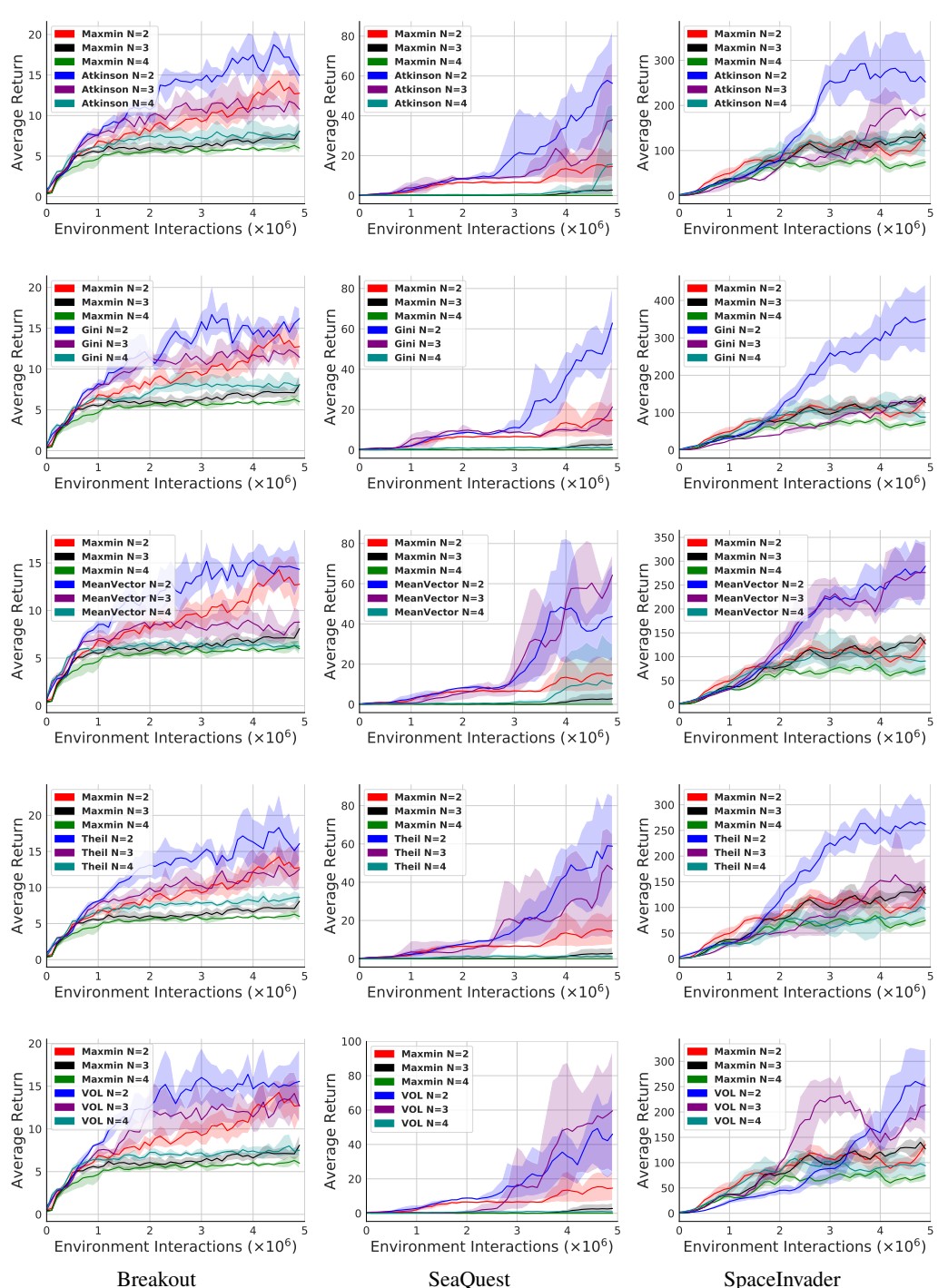

Figure 6: All MaxminDQN Results. Top to Bottom: Atkinson, Gini, MeanVector, Theil, Variance of Logarithms

## B.1 Results in Tabular Form

Table 5: Max Average Return for MED-RL MaxminDQN with two neural networks on PyGames and MinAtar environments. Maximum value for each task is bolded. $\pm$ corresponds to a single standard deviation over trials.

| Environment | Baseline | MeanVector | Gini | Atkinson | Theil | VOL |
|---|---|---|---|---|---|---|
| Asterix | 35.51 ± 8.81 | **59.20 ± 5.60** | 54.37 ± 8.32 | 52.61 ± 11.86 | 58.42 ± 13.00 | 56.82 ± 18.54 |
| Catcher | 54.72 ± 4.73 | 57.52 ± 2.31 | 57.88 ± 1.80 | 58.00 ± 1.00 | **58.58 ± 0.34** | 56.16 ± 2.96 |
| Copter | 37.83 ± 3.30 | 67.36 ± 8.23 | 68.08 ± 4.28 | 65.20 ± 5.86 | 71.80 ± 5.34 | **71.85 ± 8.64** |
| Breakout | 12.75 ± 1.67 | 14.32 ± 1.63 | **16.19 ± 1.49** | 14.93 ± 1.53 | 16.09 ± 3.05 | 15.56 ± 3.75 |
| Seaquest | 14.60 ± 9.69 | 43.67 ± 28.75 | **62.87 ± 21.68** | 56.28 ± 30.32 | 58.83 ± 26.80 | 46.04 ± 29.03 |
| SpaceInvader | 135.63 ± 14.10 | 289.64 ± 66.49 | **350.00 ± 108.27** | 252.02 ± 43.85 | 261.86 ± 29.00 | 251.64 ± 80.36 |

Table 6: Max Average Return for MED-RL MaxminDQN with three neural networks on PyGames and MinAtar environments. Maximum value for each task is bolded. $\pm$ corresponds to a single standard deviation over trials.

| Environment | Baseline | MeanVector | Gini | Atkinson | Theil | VOL |
|---|---|---|---|---|---|---|
| Asterix | 28.34 ± 4.91 | 54.17 ± 8.68 | 45.85 ± 7.6 | 47.54 ± 9.23 | 51.71 ± 9.59 | **54.50 ± 11.81** |
| Breakout | 8.10 ± 1.49 | 8.78 ± 1.38 | 11.44 ± 1.34 | 10.78 ± 2.36 | 12.54 ± 3.46 | **12.66 ± 2.49** |
| Catcher | 44.07 ± 6.10 | 58.79 ± 0.17 | **58.81 ± 0.20** | 58.68 ± 0.17 | 58.55 ± 0.61 | 57.97 ± 1.54 |
| Copter | 36.80 ± 5.75 | 68.81 ± 4.78 | **73.20 ± 3.63** | 67.55 ± 3.67 | 69.25 ± 4.00 | 65.83 ± 5.16 |
| Seaquest | 2.79 ± 3.48 | **64.28 ± 13.38** | 21.33 ± 18.43 | 38.02 ± 28.75 | 46.76 ± 25.84 | 59.59 ± 37.29 |
| SpaceInvader | 126.98 ± 18.59 | **276.24 ± 66.06** | 140.18 ± 1.52 | 180.30 ± 20.80 | 140.32 ± 40.36 | 213.85 ± 59.75 |

Table 7: Max Average Return for MED-RL MaxminDQN with four neural networks on PyGames and MinAtar environments. Maximum value for each task is bolded. $\pm$ corresponds to a single standard deviation over trials.

| Environment | Baseline | MeanVector | Gini | Atkinson | Theil | VOL |
|---|---|---|---|---|---|---|
| Asterix | 28.60 ± 5.51 | 34.16 ± 8.37 | 35.64 ± 4.03 | **41.44 ± 6.44** | 41.39 ± 10.76 | 39.35 ± 5.87 |
| Breakout | 5.98 ± 0.93 | 6.22 ± 0.68 | 8.01 ± 0.97 | 8.01 ± 1.49 | **8.64 ± 0.71** | 7.51 ± 0.81 |
| Catcher | 53.17 ± 3.24 | 58.71 ± 0.14 | 58.64 ± 0.06 | 58.83 ± 0.17 | 58.80 ± 0.06 | **58.87 ± 0.08** |
| Copter | 43.20 ± 3.06 | 68.62 ± 3.45 | 66.40 ± 6.89 | 67.18 ± 6.51 | **74.83 ± 4.71** | 63.33 ± 4.79 |
| Seaquest | 0.00 ± 0.00 | 10.26 ± 19.72 | 1.16 ± 0.75 | **15.67 ± 30.06** | 1.03 ± 0.65 | 0.87 ± 0.45 |
| SpaceInvader | 74.69 ± 10.36 | 91.09 ± 36.08 | 87.63 ± 18.68 | **120.25 ± 43.34** | 96.54 ± 16.24 | 93.89 ± 26.17 |

Figure 6 shows the training curves for the all the six environments and Tables 5 to 7 represent the results in the tabular, similar to our observations for the continuous control experiments, MED-RL augmented MaxminDQN outperformed un-regularized MaxminDQN significantly on average return and sample-efficiency metrics. Notably, in Seaquest and SpaceInvader environments where MED-RL MaxminDQN with two neural networks achieved $400\%$ and $300\%$ increase in performance. Note that the baselines shown in Figure 6 are *strong* baselines. For example, in the MaxminDQN paper, the best performance of baseline MaxminDQN on Asterix and SpaceInvader environments is around 20 and 50 respectively while we have achieved an average reward of 35 and 135 respectively.

# C    RESULTS ON DISCRETE CONTROL TASKS USING ENSEMBLEDQN

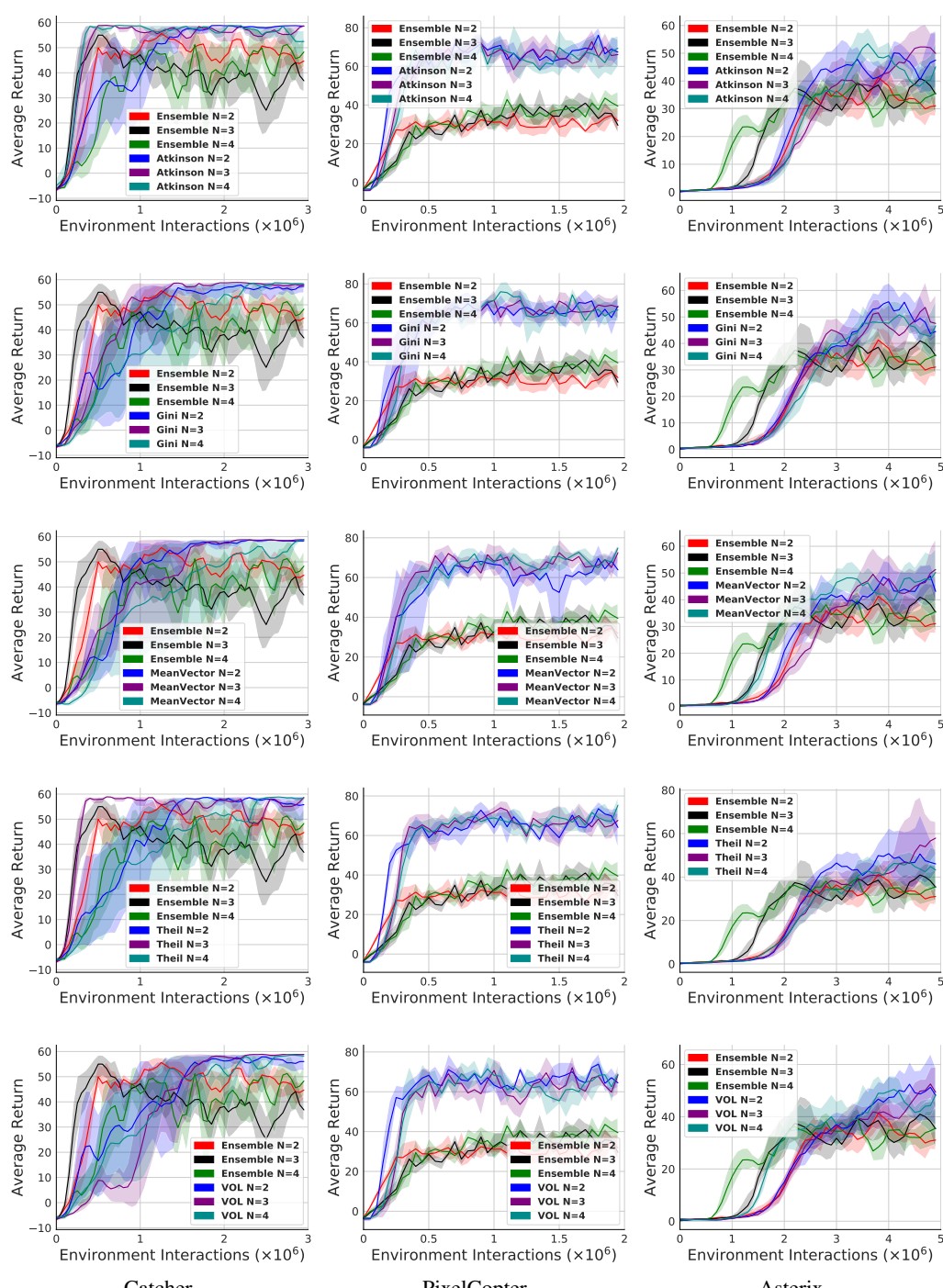

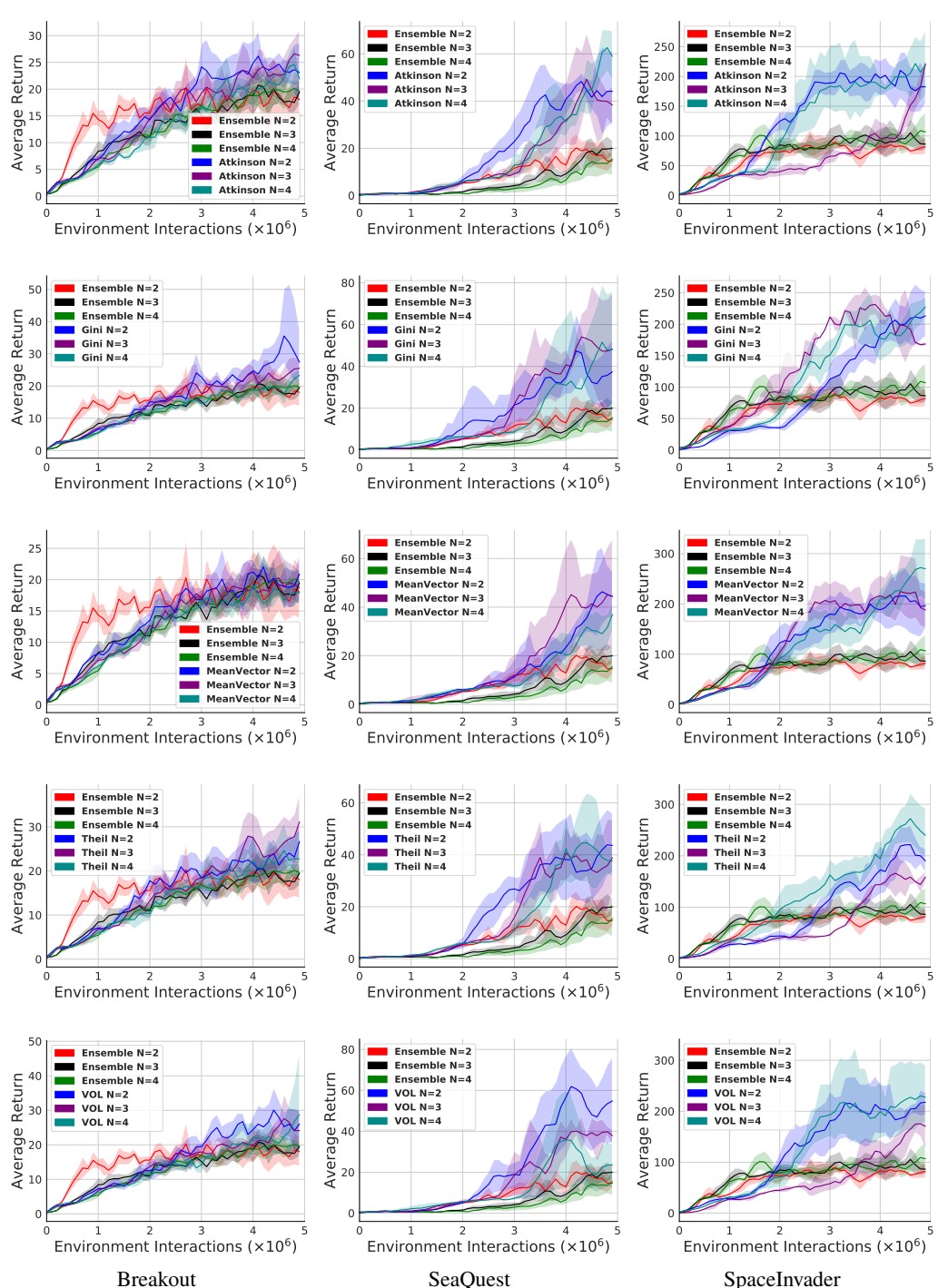

Figure 7: All EnsembleDQN Results. Top to Bottom: Atkinson, Gini, MeanVector, Theil, Variance of Logarithms

## C.1 RESULTS IN TABULAR FORM

Table 8: Max Average Return for MED-RL EnsembleDQN with two neural networks on PyGames and MinAtar environments. Maximum value for each task is bolded. $\pm$ corresponds to a single standard deviation over trials.

| Environment | Baseline | MeanVector | Gini | Atkinson | Theil | VOL |
|---|---|---|---|---|---|---|
| Asterix | $31.06 \pm 4.19$ | $43.07 \pm 4.96$ | $44.49 \pm 2.61$ | $47.52 \pm 11.76$ | $46.05 \pm 5.37$ | $\mathbf{48.21 \pm 12.68}$ |
| Catcher | $44.66 \pm 4.33$ | $\mathbf{58.29 \pm 0.70}$ | $57.29 \pm 2.60$ | $58.59 \pm 0.25$ | $55.87 \pm 5.94$ | $56.03 \pm 3.64$ |
| Copter | $31.03 \pm 8.83$ | $64.50 \pm 4.93$ | $\mathbf{69.28 \pm 6.54}$ | $67.45 \pm 7.81$ | $64.66 \pm 12.40$ | $64.03 \pm 2.37$ |
| Breakout | $18.03 \pm 4.95$ | $20.91 \pm 3.28$ | $\mathbf{27.38 \pm 10.35}$ | $23.08 \pm 6.59$ | $26.70 \pm 5.22$ | $26.05 \pm 4.38$ |
| Seaquest | $14.64 \pm 2.63$ | $44.38 \pm 10.52$ | $37.45 \pm 18.67$ | $44.17 \pm 17.62$ | $43.60 \pm 13.05$ | $\mathbf{54.84 \pm 22.53}$ |
| SpaceInvader | $81.97 \pm 15.62$ | $196.34 \pm 51.71$ | $213.32 \pm 38.77$ | $182.58 \pm 31.34$ | $190.08 \pm 27.03$ | $\mathbf{217.76 \pm 23.80}$ |

Table 9: Max Average Return for MED-RL EnsembleDQN with three neural networks on PyGames and MinAtar environments. Maximum value for each task is bolded. $\pm$ corresponds to a single standard deviation over trials.

| Environment | Baseline | MeanVector | Gini | Atkinson | Theil | VOL |
|---|---|---|---|---|---|---|
| Asterix | $35.32 \pm 10.66$ | $\mathbf{51.32 \pm 13.39}$ | $47.56 \pm 12.45$ | $49.91 \pm 8.05$ | $57.92 \pm 8.50$ | $49.37 \pm 9.78$ |
| Breakout | $19.51 \pm 2.87$ | $18.74 \pm 3.13$ | $25.52 \pm 2.81$ | $26.30 \pm 5.63$ | $\mathbf{31.15 \pm 8.19}$ | $24.14 \pm 6.66$ |
| Catcher | $37.07 \pm 5.24$ | $58.80 \pm 0.17$ | $57.95 \pm 1.16$ | $58.65 \pm 0.33$ | $58.30 \pm 0.80$ | $\mathbf{58.83 \pm 0.06}$ |
| Copter | $30.01 \pm 4.92$ | $\mathbf{73.60 \pm 4.38}$ | $67.22 \pm 8.10$ | $66.44 \pm 0.84$ | $68.12 \pm 5.93$ | $69.81 \pm 7.48$ |
| Seaquest | $19.99 \pm 3.60$ | $44.55 \pm 25.23$ | $\mathbf{48.13 \pm 25.68}$ | $38.31 \pm 6.98$ | $38.89 \pm 14.30$ | $37.84 \pm 3.41$ |
| SpaceInvader | $86.33 \pm 12.05$ | $187.88 \pm 42.91$ | $168.60 \pm 3.60$ | $\mathbf{221.38 \pm 8.48}$ | $158.65 \pm 10.45$ | $170.43 \pm 37.19$ |

Table 10: Max Average Return for MED-RL EnsembleDQN with four neural networks on PyGames and MinAtar environments. Maximum value for each task is bolded. $\pm$ corresponds to a single standard deviation over trials.

| Environment | Baseline | MeanVector | Gini | Atkinson | Theil | VOL |
|---|---|---|---|---|---|---|
| Asterix | $35.62 \pm 7.80$ | $\mathbf{50.26 \pm 11.14}$ | $46.65 \pm 9.76$ | $45.12 \pm 9.86$ | $43.23 \pm 11.50$ | $48.26 \pm 4.83$ |
| Breakout | $19.61 \pm 2.31$ | $17.85 \pm 1.82$ | $23.31 \pm 2.87$ | $21.91 \pm 1.40$ | $22.71 \pm 6.72$ | $\mathbf{28.79 \pm 18.82}$ |
| Catcher | $47.77 \pm 6.62$ | $\mathbf{58.74 \pm 0.22}$ | $58.33 \pm 0.81$ | $52.44 \pm 5.33$ | $58.24 \pm 1.11$ | $58.23 \pm 0.81$ |
| Copter | $39.29 \pm 9.75$ | $75.37 \pm 2.35$ | $67.63 \pm 6.01$ | $67.73 \pm 5.21$ | $\mathbf{75.57 \pm 5.79}$ | $70.16 \pm 3.58$ |
| Seaquest | $15.46 \pm 8.11$ | $37.00 \pm 7.16$ | $48.18 \pm 33.34$ | $\mathbf{58.95 \pm 12.31}$ | $23.59 \pm 12.35$ | $37.60 \pm 6.73$ |
| SpaceInvader | $106.76 \pm 32.14$ | $\mathbf{269.96 \pm 71.46}$ | $227.60 \pm 20.75$ | $218.65 \pm 54.32$ | $239.84 \pm 54.58$ | $227.46 \pm 63.46$ |

Figure 7 shows the training curves for the all the six environments and Tables 8 to 10 represent the results in the tabular, similar to our observations for the continuous control experiments, MED-RL augmented EnsembleDQN outperformed un-regularized EnsembleDQN significantly on average return and sample-efficiency metrics. Notably, in Copter, SeaQuest and SpaceInvader environments where MED-RL EnsembleDQN with two neural networks achieved $200\%, 350\%$ and $250\%$ increase in performance.

# D  T-SNE VISUALIZATIONS

To visualize the impact of the regularization, Figures 8 and 9 shows t-SNE (van der Maaten & Hinton, 2008) visualization of the activations of the last layer of the trained networks. Figure 8a show the network trained for the Catcher environment, while Figures 8b and 9, the network trained for the PixelCopter environment. The upper row of the figure shows the original, unregularized models, while the lower row a regularized version. For all combinations, we find that the activations from the original MaxminDQN and EnsembleDQN versions do not show any obvious pattern, while the regularized ones show distinct clusters. An additional benefit of t-SNE visualizations over CKA similarity heatmaps is that the CKA similarity heatmaps are useful to show representation similarity between two neural networks, but they become counter intuitive as the number of neural networks increases.

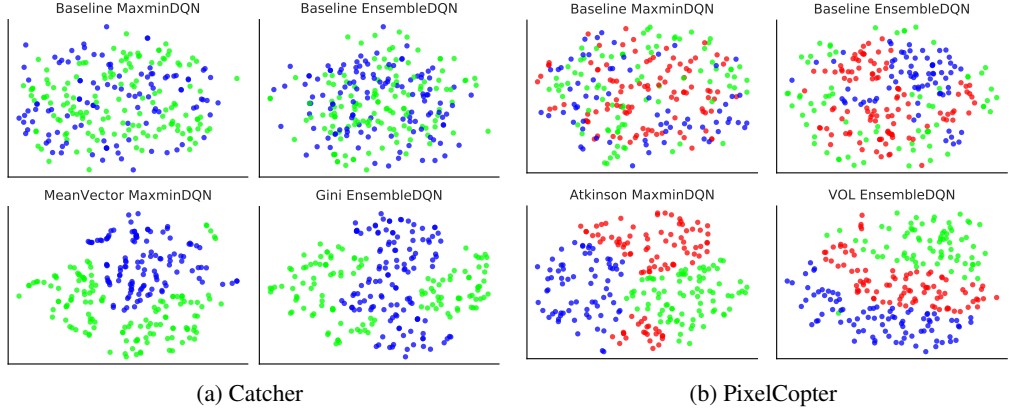

(a) Catcher                 (b) PixelCopter

Figure 8: Clustering last layer activations from Catcher and PixelCopter after processing them with t-SNE to map them in 2D. The regularized variants have visible clusters while the baseline MaxminDQN and EnsembleDQN activations are mixed together with no visible pattern.

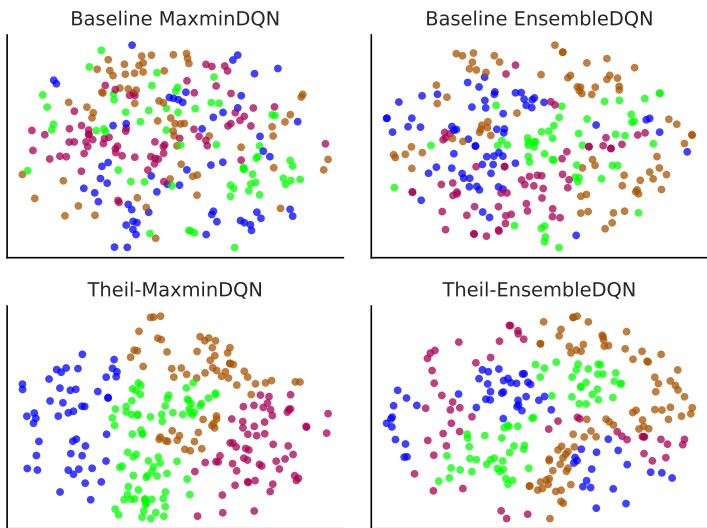

Figure 9: Clustering last layer activations from PixelCopter after processing them witht-SNE to map them in 2D

# E PLOTTING THE GINI INEQUALITY

We measured the $L^2$ norm inequality of the baseline MaxminDQN and EnsembleDQN along with their regularized versions. We trained baseline MaxminDQN and EnsembleDQN with two neural networks along with their Gini index versions with regularization weight of $10^{-8}$ on the PixelCopter environment on a fixed seed . Figure 10 represents the $L^2$ norm inequality of the experiments along their average return during training. Notably, despite each neural network being trained on a different batch, the $L^2$ norm of the baseline MaxminDQN and EnsembleDQN are quite similar while the $L^2$ norm of the regularized MaxminDQN and EnsembleDQN have high inequality.

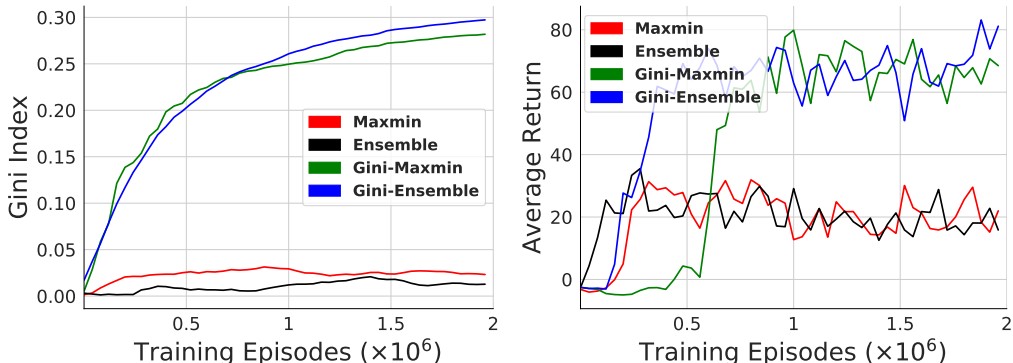

Figure 10: **Left**: Plot representing the $L^2$ norm inequality between the two neural networks using Gini index trained on PixelCopter environment. **Right**: Plot representing the average return during training.

## F  IMPLEMENTATION DETAILS AND HYPERPARAMETERS

For our implementation of MaxminDQN and EnsembleDQN, we used the code provided by the MaxminDQN authors that has implementations of different DQN based methods github.com/qlan3/Explorer). For the baseline experiments, we used most of the hyperparameter settings provided in the configuration files by the authors except the number of ensembles which we limited to four.

Table 11: Hyperparameters for discrete control tasks

| Hyperparameter | Value |
|---|---|
| Target Weight $\tau$ | $1e^{-3}$ |
| Actor Learning Rate | $[1e^{-3}, 1e^{-4}]$ |
| Regularization Weight | $1e^{-6}, 1e^{-7}, 1e^{-8}$ |
| Replay Buffer | $1e^{6}$ |
| Batch Size | 32 |
| Exploration Steps | 5000 |
| Optimizer | Adam |

Table 12: Hyperparameters for continuous control tasks

| Hyperparameter | Value |
|---|---|
| Target Weight $\tau$ | $1e^{-3}$ |
| Actor Learning Rate | $[1e^{-4}, 3e^{-5}]$ |
| Critic Learning Rate | $[1e^{-4}, 3e^{-5}]$ |
| Replay Buffer | $1e^{6}$ |
| Batch Size | $[256]$ |
| Exploration Steps | 25000 |
| Optimizer | Adam |
| Hidden Layer Size | 256 |
| Number of critics (REDQ) | 10 |
| Regularization Weight | $1e^{-6}$ |

### F.1 Computing Infrastructure

All the experiments were performed on a Kubernetes managed cluster with Nvidia V100 GPUs and Intel Skylake CPUs. Each experiment was run as an individual Kubernetes job with 11 CPUs, 16GB of RAM and 1 GPU (if needed). This configuration allowed us to run experiments without any interference from other applications which was important to accurately measure the wall-clock time.

**Future Work:** Even though the focus of this paper was on empirical testing, there are several different research questions that needs to be addressed, for example how do we select the best regularizer for a particular environment or an algorithm with some $N$ numbers of networks in the ensemble. Another interesting line of research is to study how different inequality distributions effect the diversity of the ensembles.

