# OpenReview forum: "Maximizing Ensemble Diversity in Deep Reinforcement Learning"
_ICLR.cc/2022/Conference — ICLR 2022 Poster_

### Official Review · Reviewer_TfGq · 2021-10-28

**Correctness:** 2
**Technical Novelty And Significance:** 2
**Empirical Novelty And Significance:** 3
**Recommendation:** 6
**Confidence:** 4

**Main Review:**

strengths:
+ A simple but general regularization method to improve the performance of Q-function ensemble-based RL methods is proposed.
As I mentioned in the summary of the paper, a method to regularize the similarity of the parameters of the Q-functions is proposed in the paper.
The similarity of parameters between Q functions is evaluated based on five measures (e.g. Gini coefficients), and then added to the loss for the Q functions.
Since we only need to add a similarity score to the loss, the proposed method can be easily introduced into the existing ensemble-based RL methods.
+ The proposed regularization method is shown to be effective in various RL methods and environments.
The proposed regularization method is shown to be effective in five different RL methods (SAC, TD3, REDQ, Maxmin-DQN, and Ensemble-DQN).
In addition, the regularization method is evaluated in continuous control (6 Mujoco envs.) and discrete control (6 Atari envs.).



weaknesses:

- There is no theoretical justification for introducing the regularization method based on the similarity measures.
  - For example, it is not clear if Q-functions converge to the optimal one (under reasonable assumptions) when they are trained with the regularized method.
  - Also, there is no theoretical discussion of how the use of similarity measures affect overall performance improvement.
  - Nevertheless, I regard the study presented in the paper as an empirical one, and thus do not think that the lack of theoretical justification is a severe weakness of the paper.

- The quality of the presentation is not sufficiently good:
  - Connections among the sections in the paper are somewhat unclear.
    - For example, in Sections 3 and 4.1, the similarity of the Q functions (neural nets) is measured by the similarity of their activation units' outputs. However, in Section 4.2 and later, the similarity of Q functions is measured by the similarity of their parameter values. In sum, the criterion used for evaluating similarity differs among the sections.
    - In addition, Algorithm 1 in Section 4 describes the Maxmin-DQN variant that uses the proposed regularization method. However, this algorithm (and Maxmin-DQN) do not appear in the later sections (except Appendix).

  - Experiment setups are not clearly explained. In particular, the setup in Section 5.4 is very unclear, e.g.,:
    - Steps taken by MED-RL to reach REDQ performance:  How this number of steps is counted? How MED-RL's performance is evaluated? (is it MED-RL's return in a single test episode? or its average return over multiple test episodes?)
    - Wall clock time REDQ (in mins): Is this the time for REDQ to complete 300k interactions with an environment? If so, what if REDQ's performance converges before 300k interactions?
    - REDQ baseline: Is the REDQ's learning curve the one shown in Figure 5? If so, why is it much worse than the one shown in the original REDQ paper?

Minor comment:
The figures in the pdf of the paper (e.g. Figures 3 and 4) take a very long time to be rendered.
This is probably because all the data points in the graphs are output in vector format.
If you want to put graphs with many data points on the same page, it is better to rasterize the graph.



**Summary Of The Paper:**

In this paper, it is proposed to regularize the parameters of each Q function in ensemble so that they are not similar to each other. Through various experiments, it is shown that RL methods that use ensembles of Q-functions (e.g., REDQ, Minmax-DQN, etc) can improve its sample efficiency by using the regularization.

**Summary Of The Review:**

I'm leaning to recommend reject.
I acknowledge the significance of the finding that it is useful to keep the divergence among the Q-function.
I also acknowledge the authors' effort to demonstrate the usefulness of the regularization method in various RL methods and environments.
However, the clarity of the current version of the paper does not meet the threshold for publication, and a non-trivial revision is needed.

----- Update after author's revision 20211125 ------------------------------
The authors have improved the explanation of experimental setups, but the connections between the sections still have not been sufficiently improved.
In particular, some of the other reviewers are concerned about the mismatch between the similarity criterion (CKA) used in Section 4.1 and the similarity criterion used in later sections.
I have seen their discussions with the authors, and think that their concern about the mismatch is not sufficiently resolved.
I think that replacing the analysis based on CKA with an analysis using criteria based on parameter values (e.g., equation (4)) would make the discussion in the paper more consistent.

Overall, I acknowledge that the paper was improved to some extent in the revision, and thus slightly improve the score (WR->WA).
However, as mentioned above, I have still concerns about the clarity of the paper and cannot strongly champion the paper to accept.

---

> ### Author Response · Authors · 2021-11-17
> **Rebuttal**
>
> Thank you so much for your review and especially for providing valuable feedback to enhance the organization of the paper which we have incorporated in the paper.
>
> To address the weaknesses pointed out in the review, we do agree that this paper lacks theoretical discussion but as we have mentioned that the nature of the paper is empirical therefore, we compensated the lack of theoretical discussion with a large number of experiments across both continuous and discrete domains and on a wide variety of ensemble settings.
>
> Q: How is this number of steps counted?
>
> Ans: The “steps” mentioned in Table 4 refers to the environment interactions. We have already corrected this in the table.
>
> Q: How is MED-RL's performance evaluated? (is it MED-RL's return in a single test episode? or its average return over multiple test episodes?)
>
> Ans: Since we log test runs after 10K environment interactions along with timestamps, for every seed, we look at the environment interactions count at a point where the MED-RL’s performance exceeds the average performance of the REDQ mentioned in table 3 and yes, it is the average return over multiple test  seeds.
>
> Q: Is this the time for REDQ to complete 300k interactions with an environment? If so, what if REDQ's performance converges before 300k interactions?
>
> Ans: This is a great question but if we look at Figure 5, we can see that apart from the Hopper environment, the highest performance achieved by REDQ is towards the end of the training cycle near 300K environment interactions. Therefore, the  results shown in Table 4 still hold true.
>
> Q: REDQ baseline: Is the REDQ's learning curve the one shown in Figure 5? If so, why is it much worse than the one shown in the original REDQ paper?
>
> This is true that the plots shown in Figure 5 represent  the results from REDQ. Since we have used the code provided by the authors, this discrepancy could possibly be because of the random seeds we have used (we used a fixed set of random seeds across all continuous control experiments).
>
> But we would also like to point your attention to results shown in Figure 6. The baseline results for MaxminDQN specially for Asterix and SpaceInvader are up to twice as high as reported in the original paper. Similarly, for SAC, the reported baseline results in our paper match or exceed (Ant and Walker) the results reported in the original paper.
>
> Also, we will update the paper with compressed figures and recommended  changes soon.

---

> > ### Comment · Reviewer_6miY · 2021-11-17
> > **As for the performance difference of original implementations and reported results**
> >
> > Thanks a lot for the reply!
> >
> > > This is true that the plots shown in Figure 5 represent the results from REDQ. Since we have used the code provided by the authors, this discrepancy could possibly be because of the random seeds we have used (we used a fixed set of random seeds across all continuous control experiments).
> >
> > > But we would also like to point your attention to results shown in Figure 6. The baseline results for MaxminDQN specially for Asterix and SpaceInvader are up to twice as high as reported in the original paper. Similarly, for SAC, the reported baseline results in our paper match or exceed (Ant and Walker) the results reported in the original paper.
> >
> > As for this, doesn't this imply that the number of seeds (5) is insufficient to draw a meaningful conclusion? For example, MaxminDQN paper seems to use 20 seeds while this paper use only 5 seeds (if I understand correctly). Also, can you run REDQ with original seeds? Maybe released REDQ open source contains a bug or some difference from the original implementation... We still have time to confirm this, I think.
> >
> > As for SAC comparison, the original paper uses v1 envs, but I guess you used v3 envs. There are some changes between v1 and v3, so they are not really comparable. SpinningUp's benchmark is performed on v3 envs, and probably the results in the paper should be compared against SpinningUp's benchmark. As far as I quickly compared SAC results in this paper against SpinningUp's benchmark, it seems the results in this paper are generally worse except Ant.
> >
> > That being said, since REDQ is the state-of-the-art of ensemble-based methods, I think it would be nice to run REDQ and the proposed method with more seeds so that we can clearly see the proposed method beats REDQ. SAC is somewhat irrelevant, and REDQ beats maxminDQN, as far as I understand.

---

> > > ### Author Response · Authors · 2021-11-18
> > > **Thank you for the response**
> > >
> > > Thank you so much for the response, we just want to mention that the original REDQ paper has performed experiments on 5 random seeds but we will try to add more experiments (if not all) till the deadline day.

---

> > > > ### Comment · Reviewer_6miY · 2021-11-18
> > > > **Thanks a lot for additional experiments**
> > > >
> > > > > we just want to mention that the original REDQ paper has performed experiments on 5 random seeds
> > > >
> > > > Exactly. So, it is possible that the high performance reported in the original paper turns out to be just due to nice random seeds.
> > > >
> > > > Anyway, thanks a lot for additional experiments. I am looking forward to seeing their results. While some baseline results are worse than previously reported results, the use of ensemble-inducing regularizations seem to consistently improve original algorithms. I suggest the additional experiments to corroborate the claims and strengthen the paper.

---

> > ### Comment · Reviewer_TfGq · 2021-11-27
> > **Thank you for your response**
> >
> > I have updated my review comments (see "Update after author's revision 20211125" in "Summary Of The Review:").

---

### Official Review · Reviewer_6miY · 2021-11-01

**Correctness:** 2
**Technical Novelty And Significance:** 1
**Empirical Novelty And Significance:** 3
**Recommendation:** 6
**Confidence:** 3

**Main Review:**

# Strong Points of the Paper

- S1. It is nice that one can gain a considerable sample efficiency and performance improvement by such a simple technique.
- S2. It is intriguing that the performance of ensemble-based deep RL methods highly (negatively) correlates with the representation similarly among ensembles of neural networks.
- S3. I highly appreciate that the paper provides many experimental results with different regularizers, baseline algorithms, and the number of ensembles.

# Weak Points of the Paper

I give detailed comments about these points in the next section.

- W1. From Figure 1, I am not so convinced that the performance of ensemble-based deep RL methods highly (negatively) correlates with the representation similarity.
- W2. It is unclear how hyper-parameters and seeds are chosen.
- W3. Some comparison seems to be unfair.

# Detailed Comments

As W1 above, what is meant by "high" and "low" CKA similarity? In heatmaps at point A and C, (near-)diagonal elements have low values, but other elements typically have higher values. In contrast, heatmaps at point B and D show the opposite tendency, and it is not so convincing to me that there is a negative correlation between performance and layer x-y similarity. Furthermore, only showing results at 4 points is not convincing to me. I trust the authors, but it is known that there is an implicit bias that causes cherry-picking without any intention. Can you plot curves of layer x-y similarity along with the performance?

As for W2 above, how did you choose hyperparameters and seeds? There are some places where the authors say "seeds are fixed". However, does this mean that you fixed random seeds, ran experiments with different hyperparameters, and pick up the best results? If you did this, there may be a maximization bias. In addition, running experiments with only 5 seeds is probably insufficient (Henderson et al. 2018). Please run experiments with more random seeds.

As for W3, it seems to me that REDQ's performance in this paper is significantly lower than that of original paper. Since the authors of REDQ's paper opensourced their code [here](https://github.com/watchernyu/REDQ), I don't think reporting re-implemeted REDQ's result with a lower performance is fair. (Or you use it, and I just missed that?)

# Questions

- Q1. As for proposed indices, it seems to me that you can easily maximize some (maybe all?) of them, especially when ReLU is used. For example, in case of Gini coefficient, you can multiply weights of the second to the last layer by a huge constant, and then, multiply the weights of the last layer by a small constant. By choosing the constants carefully, you can get as high $L^2$ norm as you like without affecting the output and causing a high Gini index. Why doesn't this occur? Maybe I am missing something?
- Q2. Why don't you simply minimize the sum of CKA? It is possible to backpropagate through it, right? Figure 1 indicates that low CKA implies high performance, so it is a very natural idea to minimize the sum of CKA.

# Minor Comment

Would you replace Figure 2(a) with a lighter one? It takes a long time to load the figure on my PC, and I can't print out the PDF probably because of it. I guess it is because the figure is a vector format, and there are too many points in the figure. Maybe replacing it with JPG or PNG would resolve this issue.

# References

Deep Reinforcement Learning that Matters; Peter Henderson, Riashat Islam, Philip Bachman, Joelle Pineau, Doina Precup, David Meger. AAAI 2018


**Summary Of The Paper:**

The paper considers a problem of ensemble-based deep RL methods that ensembles of critic networks converge to the same point in the representation space. To address it, the paper proposes a regularization technique that forces representations of a critic network to be dissimilar from those of other critic networks in the ensemble. It is empirically shown that this regularization technique improves the sample efficiency and asymptotic performance of baseline algorithms.

**Summary Of The Review:**

Figure 1, which is intended to demonstrates the negative correlation between representation similarity and performance, is not convincing to me. Furthermore, the experiment conditions seem to be insufficient or not explained well.

---

> ### Author Response · Authors · 2021-11-18
> **Response**
>
> Thank you so much for the review. As per your request and discussion below, we  are running a few more experiments on the REDQ  and if time permits, on the other algorithms as well.
>
> Q: multiply the weights of the last layer by a small constant?
>
> Ans: As far as we understand this question, does this mean to do the aforementioned trick at the initialization time or during training? If we do this during the initialization, this would be similar to any other random initialization.
> Our other understanding is that this method suggests to measure inequality on the layer level which we are not doing, we are trying to maximize inequality on the whole neural network level. Further clarification would be great.
>
> Q: Why don't you simply minimize the sum of CKA?
>
> Ans: As we mentioned above, There are primarily two reasons we haven’t used the CKA metric directly, which seems to be the obvious choice. The first reason is that although we want to maximize diversity in the neural nets, at some point during the training we want neural networks to output the same value when they reach the optimal Q-values. For example, MaxminDQN addresses the overestimation bias problem, if we try to minimize the CKA directly, over time, the variance between the Q-values will have enough variance to prevent it from learning the optimal policy or might suffer from a catastrophic collapse.
>
> The second reason for not choosing the CKA metric is purely computational. As shown in section 3, CKA is a kernel based metric, the computation time to calculate CKA increases with batch size. Although we have kept the batch size small, some applications might need a fairly large batch size to train properly. Additionally, CKA measures similarity between two neural networks. Using CKA might be feasible when the number of ensembles is small but for a large number of ensembles it can be prohibitively expensive. For example, REDQ uses 10 networks, at every gradient step, we would have to calculate CKA 10 choose 2 times and this number can grow as the number of networks increases. Therefore, for all the reasons mentioned, we chose the L2 norm with economics inspired metrics to maximize the diversity between the neural networks.
>
> Q: what is meant by "high" and "low" CKA similarity?
>
> Ans: The term CKA similarity means how much the representation of a neural network is similar to that of another neural network. Since we have identical architecture for our experiments, the most relevant information on the heatmaps is the cross diagonal (from bottom left to top right). The values of the cross diagonal represents similarity between two corresponding layers of the network, for example, the value of the bottom left shows how similar the output of the first layers of both neural networks are.

---

> > ### Comment · Reviewer_6miY · 2021-11-24
> > **Re: Response**
> >
> > Thanks a lot for the response.
> >
> > > Q: multiply the weights of the last layer by a small constant?
> >
> > Sorry for insufficient explanation. For example, let's consider a 1-hidden layer NN with only one neuron in each layer. Then, the output $y$ is $y = w_{last} relu(w_{first} x)$, where $x$ is the input. Since $relu(v) = \max (v, 0)$, if we multiply $w_{last}$ by, say $c \in (0, \infty)$, and $w_{first}$ by $1/c$, the output is $c w_{last} relu(w_{first} x / c) = w_{last} relu(w_{first} x) = y$. However, $l_2$ norm of weights changes, and Gini coefficient seems to increases. In particular, when $c \rightarrow \infty$, it is maximized, I guess.
> >
> > > Q: Why don't you simply minimize the sum of CKA?
> > > Q: what is meant by "high" and "low" CKA similarity?
> >
> > I got it. Maybe better to explain in the paper.

---

> ### Author Response · Authors · 2021-11-23
> **Update: More Experiments**
>
> Thank you again for your feedback. As per your feedback, we have added  more results  (total 8 seeds) for REDQ and SAC (Appendix G and H) for now. Similar to results before, MED-RL augmented  REDQ and SAC both outperformed baseline REDQ and SAC.

---

> > ### Comment · Reviewer_6miY · 2021-11-24
> > **Re: Update: More Experiments**
> >
> > Thanks a lot for additional runs. Just let me confirm one minor thing: did you use seeds the original REDQ implementation used? If REDQ's performance is still poor with those seeds, the open-sourced code probably contains a bug.

---

> > > ### Author Response · Authors · 2021-11-24
> > > **Response**
> > >
> > > We have used their function to set the seed. Also, we are using the exact codebase to add diversity.

---

> > > > ### Comment · Reviewer_6miY · 2021-11-24
> > > > **Re: Response**
> > > >
> > > > I see. Thank you for the confirmation.

---

### Official Review · Reviewer_a9sA · 2021-11-02

**Correctness:** 3
**Technical Novelty And Significance:** 3
**Empirical Novelty And Significance:** 3
**Recommendation:** 6
**Confidence:** 3

**Main Review:**

This paper is clearly written and easy to follow. I'm always happy to see simple effective techniques that can substantially improve the performances of existing methods in challenging testbeds, which looks promising to me.

**Question**
My biggest question is that all the criteria are evaluated w.r.t. the l2 norm of the whole network parameters, which is very counter-intuitive to me. Note that the paper simply uses the CKA metric to evaluate layer similarities for the purpose of visualization and interpretation. Wouldn't it be a more natural choice to use some metrics that at least involve CKA or layer-wise information? For example, can we simply optimize CKA directly in the pseudo-code instead of using $\mathcal{I}(l_i,l)$ as the auxiliary objective? Could the authors provide some in-depth analysis and justifications on why simply the l2 norm is preferred? Or, is it possible to provide some more experiments that use CKA directly?

**Possible Improvement**
From the additional diversity training objective highlighted in the pseudo-code, it is conceptually related to the diversity learning literature in population-based-training, where a population (corresponding to ensemble to some extent) of policies are trained and each policy not only optimizes its reward objective but also optimizes a diversity metrics for the purpose of effectively covering diverse strategic modes. So it would be appreciated if the authors can provide some discussions in the related work section. Some of the references are listed here for the purpose of helping the authors to survey the literature more easily: https://arxiv.org/abs/2002.00632, http://proceedings.mlr.press/v139/lupu21a.html, https://arxiv.org/abs/2103.04564, https://arxiv.org/abs/2106.02195

**Summary Of The Paper:**

This paper proposes a collection of diversity metrics to improve ensemble diversity, which substantially improves learning efficiency for a variety of RL methods.

**Summary Of The Review:**

In general, although the paper can be still improved, the general results and suggested techniques are promising.

---

> ### Author Response · Authors · 2021-11-16
> **Rebuttal**
>
> Thank you so much for your review and for bringing to our attention the related work about diversity in reinforcement learning and have added it to the related work section.
>
> Q: Wouldn't it be a more natural choice to use some metrics that at least involve CKA or layer-wise information?
>
> Ans: There are primarily two reasons we haven’t used the CKA metric directly, which seems to be the obvious choice. The first reason is that although we want to maximize diversity in the neural nets, at some point during the training we want neural networks to output the same value when they reach the optimal Q-values. For example, MaxminDQN addresses the overestimation bias problem, if we try to minimize the CKA directly, over time, the variance between the Q-values will have enough variance to prevent it from learning the optimal policy or might suffer from a catastrophic collapse.
>
> The second reason for not choosing the CKA metric is purely computational. As shown in section 3, CKA is a kernel based metric, the computation time to calculate CKA increases with batch size. Although we have kept the batch size small, some applications might need a fairly large batch size to train properly. Additionally, CKA measures similarity between two neural networks. Using CKA might be feasible when the number of ensembles is small but for a large number of ensembles it can be prohibitively expensive. For example, REDQ uses 10 networks, at every gradient step, we would have to calculate CKA 10 choose 2 times and this number can grow as the number of networks increases. Therefore, for all the reasons mentioned, we chose the L2 norm with economics inspired metrics to maximize the diversity between the neural networks.

---

### Official Review · Reviewer_i4M1 · 2021-11-04

**Correctness:** 3
**Technical Novelty And Significance:** 2
**Empirical Novelty And Significance:** 2
**Recommendation:** 3
**Confidence:** 4

**Main Review:**


The claim in abstract that “members of the ensemble can converge to the same point either the parametric space or representation space during the training phase “. This is not true for DRL. Neural networks converge to different solutions given the initialization is different and multiple local minima.

Questions:
1. Why these environments were chosen?

2. I get your Sec. 4.1. However, this is far from compelling. The claim of your conjecture is pretty big and here you have a small experiment about only one algorithm at only two time points.

3. In Alg 1, what is I()? Is it a general function that you have a few alternatives in Sec 4?

4. Sec 5.1: how do you initialize the networks in reaching the conclusion that “each neural network is trained on a separate batch from the replay buffer but still learns similar representations. “. Did you try another initialization method? How come the cross-diagonal of the after-training not exactly being ones?  Is the CKA non-symmetric? Why do you say the similarity is high? 0.15 between layer 6 and 7 is pretty low.

Ensembles in Deep RL: Most the cited papers used ensembles for the critic (Q function). There is also ACE algorithm that uses ensembles from actors. Building an ensemble from actors can help agents explore at the option level.

ACE: An Actor Ensemble Algorithm for Continuous Control with Tree Search
https://arxiv.org/abs/1811.02696

The results are not clearly separatable. Many lines are hard to see their statistical importance especially only 5 runs were performed.

Minor:
Lan Qingfeng. Gym compatible games for reinforcenment learning.
The author name for this paper reversed first name and last name.


**Summary Of The Paper:**

MED-RL
This paper studies foster diversity in ensemble of DRL networks by regularization methods. The paper is an empirical one and compared five ensemble methods with and without their diversity algorithm in six Mujoco and six Atari games, and showed some results.
The algorithm proposed is a modification of MaxMinDQN by Lan et.al. with a regularization.

**Summary Of The Review:**

The paper had a claim that initialization of different networks is not effective in proving diversity. This is not well supported. It can give a wrong message to the literature with the small and insufficient studies. this is my main concern.

---

> ### Author Response · Authors · 2021-11-15
> **Rebuttal**
>
> Thank you so much for your review. We would like to start by reiterating that we added diversity enhancing regularization to MaxminDQN and EnsembleDQN for the discrete control tasks and to TD3, SAC and REDQ for the continuous control tasks, not only to MaxminDQN as you have mentioned in the summary.
>
> Q: Why were these environments chosen?
>
> Ans: We believe this question is related to the MinAtar and PyGame environments that we used for discrete control tasks. As we have mentioned in page 9, since, we used the source code provided by MaxminDQN authors and they have used these exact environments for their experiments, so for fair comparison, we have used the same environments in our experiments.
>
> Q: In Alg 1, what is I()? Is it a general function that you have a few alternatives in Sec 4?
>
> Ans: Yes this is exactly true, I() can be any of the alternatives mentioned in section 4, we are going to update the document to be more clear about this.
>
> Q: Sec 5.1: how do you initialize the networks?
>
> Ans: The network layers are initialized  using the following code
>
>
>      def layer_init(layer,w_scale=1.0):
>
>
>
>
> 	#Initialize all weights and biases in layer and return it
>
>
> 	nn.init.orthogonal_(layer.weight.data)
>
>
> 	nn.init.xavier_normal_(layer.weight.data, gain=1)
>
>
> 	nn.init.kaiming_normal_(layer.weight.data, mode='fan_in', nonlinearity='relu')
>
>
> 	layer.weight.data.mul_(w_scale)
>
>
> 	nn.init.constant_(layer.bias.data, 0)
>
>
> 	return layer
>
>
> The exact same initialization function is used by the MaxminDQN authors and  we kept it the same  for section 5.1 experiments, also note that the number of neurons in each layer are also different, along with the batch size and the learning rate
>
> Q: Why do you say the similarity is high? 0.15 between layer 6 and 7 is pretty low.
>
> Ans: The 0.15 you mentioned is similarity between layer 6 of network A and layer 7 of network B, which is irrelevant for this paper. We are interested in cross diagonal values (from bottom left to top right) as mentioned in the caption of figure 2.
>
> Q: The paper had a claim that initialization of different networks is not effective in proving diversity.
>
> Ans: We respectfully disagree that we made this claim as a novelty. In section 5.1, we summarized the findings of [1] that random initialization is not a good method to induce diversity. For your ease, we quote from [1], “Starting each network with differing random initial weights will increase the probability of continuing in a different trajectory to other networks. This is perhaps the most common way of generating an ensemble, but is now generally accepted as the least effective method of achieving good diversity.”
>
> [1] Gavin Brown. Diversity in Neural Network Ensembles. PhD thesis, University of Birmingham,
> United Kingdom, 2004. Winner, British Computer Society Distinguished Dissertation Award.

---

> ### Author Response · Authors · 2021-12-06
> **Checking in**
>
> Respected Reviewer,
>
> Checking in if we have addressed your concerns in our rebuttal? Please let us know if you have any other question. Thank you so much.

---

### Decision · Program_Chairs · 2022-01-20

**Decision:**

Accept (Poster)

**Comment:**

This paper concerns ensemble methods in deep reinforcement learning, examining several such methods, and proposes to address an important issue wherein ensemble members converge on a representation of approximately the same function, either by their parameters converging to an identical point or equivalent points that give rise to the same function. The authors propose a set of regularization methods aimed at improving diversity, and benchmark these augmentations on five ensemble methods and a dozen environments.

3 of 4 reviewers generally praised the method's simplicity and generality, and found the experiments convincing. Reviewer a9sA describes it as "clearly written and easy to follow", although others found clarity lacking in parts. There was agreement among these 3 reviewers that this was an interesting problem to tackle. Reviewer TfGq notes that this method lacks theoretical justification or guarantees, but that as a largely empirical paper this is perhaps of secondary importance. Reviewers 6miY and a9sA had questions about the precise choice of metrics, hyperparameters and seeds; the resulting discussion cleared up many of these concerns.

The most critical reviewer, i4M1, disputes the existence of the phenomenon at all, saying that "Neural networks converge to different solutions given the initialization is different and multiple local minima." The remainder of i4M1's criticisms seem centered on the choice of environments and the number of seeds (also raised by other reviewers). The issue of seeds has been addressed partially and the authors have committed to strengthening their results in this regard.

Reviewer i4M1's statement on the convergence of neural networks to different minima matches a bit of dated folk wisdom about neural networks, but the AC disputes this. The authors have cited a study from before the DL era properly began that identifies this issue and Section 5 addresses these criticisms directly. In practice, modern neural networks, especially with non-saturating activations, tend to be surprisingly consistent across random seeds when trained against the same data stream, and more recent work posits that the loss landscape is less riddled with local minima than with saddle points (see e.g. Dauphin et al, 2014). _Equivalent_ minima are of course common due to scaling and permutation symmetries but SGD has a well documented preference for low norm solutions in the former case, and the authors' have chosen methods that would at least conceivably overcome these issues, by focusing on summary statistics of the representations rather than their precise values (and indeed, CKA is designed with these concerns in mind).

Despite i4M1's incredulity I am inclined to agree with the majority of reviewers and view the paper as a worthwhile contribution to the body of knowledge (purely empirical though it may be) on both NN ensemble methods and DRL ensembles in particular. The introduction of measures from economics is clever and original, and the results are promising. A more exhaustive study on the entire Atari57 benchmark but can appreciate the resource problem this poses, and find that the suite of considered environments, combined with the augmentation of 5 different DRL ensemble methods, strikes a good balance. I concur on the issue of seeds and would encourage authors to include as many as possible for the camera ready, but on balance would recommend acceptance.